# Multi-Omics Integrative Analyses Identified Two Endotypes of Hip Osteoarthritis

**DOI:** 10.3390/metabo14090480

**Published:** 2024-09-01

**Authors:** Jingyi Huang, Ming Liu, Hongwei Zhang, Guang Sun, Andrew Furey, Proton Rahman, Guangju Zhai

**Affiliations:** 1Human Genetics & Genomics, Division of BioMedical Sciences, Faculty of Medicine, Memorial University of Newfoundland, St. John’s, NL A1B 3V6, Canada; jingyih@mun.ca (J.H.); mingl@mun.ca (M.L.); 2Discipline of Medicine, Faculty of Medicine, Memorial University of Newfoundland, St. John’s, NL A1B 3V6, Canada; hzhang@mun.ca (H.Z.); gsun@mun.ca (G.S.); prahman@mun.ca (P.R.); 3Discipline of Surgery, Faculty of Medicine, Memorial University of Newfoundland, St. John’s, NL A1B 3V6, Canada; afurey@mun.ca; 4Office of the Premier, Government of Newfoundland & Labrador, St. John’s, NL A1B 4J6, Canada

**Keywords:** hip osteoarthritis, endotypes, metabolomics

## Abstract

(1) Background: Osteoarthritis (OA) is a heterogeneous disorder, and subgroup classification of OA remains elusive. The aim of our study was to identify endotypes of hip OA and investigate the altered pathways in the different endotypes. (2) Methods: Metabolomic profiling and genome-wide genotyping were performed on fasting blood. Transcriptomic profiling was performed on RNA extracted from cartilage samples. Machine learning methods were used to identify endotypes of hip OA. Pathway analysis was used to identify the altered pathways between hip endotypes and controls. GWAS was performed on each of the identified metabolites. Transcriptomic data was used to examine the expression levels of identified genes in cartilage. (3) Results: 180 hip OA patients and 120 OA-free controls were classified into three clusters based on metabolomic data. The combination of arginine, ornithine, and the average value of 7 lysophosphatidylcholines had an area under the curve (AUC) of 0.97 (95% CI: 0.96–0.99) to discriminate hip OA from controls, and the combination of γ-aminobutyric acid, spermine, aconitic acid, and succinic acid had an AUC of 0.96 (95% CI: 0.94–0.99) to distinguish two hip OA endotypes. GWAS identified 236 SNPs to be associated with identified metabolites at GWAS significance level. Pro-inflammatory cytokine levels were significantly different between two endotypes (all *p* < 0.05). (4) Conclusions: Hip OA could be classified into two distinct molecular endotypes. The primary differences between the two endotypes involve changes in pro-inflammatory factors and energy metabolism.

## 1. Introduction

Osteoarthritis (OA) is a chronic disabling disease that affects all parts of the joints. It is highly prevalent in the elderly population, and results in a substantial burden on health systems [1]. According to a recent and comprehensive meta-analysis involving 326,463 participants, the prevalence of hip OA was 8.55% globally [2]. Up to date, total joint replacement (TJR) surgery is the most effective treatment for end-stage OA. However, due to its multifactorial and heterogeneous nature, patients exhibit varied responses to TJR [3]. A systematic review reported six distinct clinical phenotypes in knee OA patients: chronic pain, inflammatory, metabolic syndrome, bone and cartilage metabolism, mechanical overload, and minimal joint disease [4]. Roemer F. et al. focused on imaging examination results and proposed four structural phenotypes in knee OA: inflammatory phenotype, subchondral bone phenotype, meniscus-cartilage phenotype, and hypertrophic and atrophic phenotypes [5]. The phenotypic diversification can significantly affect therapeutic efficacy due to different underlying aetiologies [6]. However, there is no consensus yet with regards to classifying OA patients into subgroups.

The lack of consensus in subsetting OA patients creates a barrier to the development of more targeted and effective treatments in OA patients [7]. Unlike phenotypes, which are observable physical characteristics of an organism, endotypes are disease subtypes defined by distinct pathophysiological mechanisms [8]. By linking with biochemical markers, patients can be assigned to specific subgroups for clinical stratification and personalized medicine. Angelini F. et al. suggested that knee OA can be classified into three endotypes: low tissue turnover, structural damage, and systemic inflammation, which have different clinical symptoms and pathological progressions [9]. Yuan C. et al. proposed four knee OA subtypes: glycosaminoglycan metabolic disorder subtype, collagen metabolic disorder subtype, activated sensory neuron subtype, and inflammation subtype; based on the results of high-throughput transcriptomics, gene ontology was used to link the clinical symptoms and molecular functions of the proposed OA subtypes [10]. A multi-omics study was conducted on synovial fibroblast samples from obese and non-obese OA patients using targeted proteomics, metabolomics, and transcriptomics [11]. This study observed differences in the inflammatory landscape between the two groups and identified distinct synovial fibroblast subsets in each group. Our previous work proposed three potential endotypes of OA: muscle weakness, arginine deficiency, and low-grade inflammation [12]. However, most of these studies focused on knee OA or did not differentiate affected joints. 

Furthermore, differences in bone geometry, mechanical stress patterns, OA prevalence, and OA-associated genetic factors have been observed between knee and hip joints, highlighting the necessity for joint-specific research [13]. However, data on hip OA endotypes are still sparse, which results in a lack of comprehensive understanding of the molecular mechanisms of hip OA, thus slowing the development of effective treatment strategies. Therefore, we undertook this multi-omics study to investigate the endotype of hip OA.

## 2. Materials and Methods

### 2.1. Study Participants

The current study was part of the Newfoundland Osteoarthritis Study (NFOAS) that recruited OA patients between November 2011 and September 2017 at St Clare’s Mercy Hospital and Health Sciences Centre General Hospital in St John’s, Newfoundland & Labrador (NL), Canada [12]. The diagnosis was based on the American College of Rheumatology OA clinical diagnostic criteria, and was confirmed by the attending orthopedic surgeon as well as the pathology reports of the removed articular cartilage following surgery [14]. OA-free controls were derived from an existent population-based cohort collected from the same source population as the NFOAS—the Complex Diseases in the Newfoundland Population: Environment and Genetics (CODING) study, which is an ongoing large-scale nutrigenomics study in NL, Canada [15]. Hip OA patients in the current study were defined as individuals who underwent total hip joint replacement (THR) due to primary hip OA, whereas OA-free controls were defined as individuals with no diagnosis of OA in any joints by their family practitioners. Patients who underwent bipolar arthroplasty for non-pathological femoral neck fracture and had no evidence of hip OA according to their medical records were collected as OA-free cartilage, and further confirmed by pathological examination of the resected femoral head cartilage. 

Ethics approval for these studies was received from the Health Research Ethics Authority of Newfoundland and Labrador (HREB #10.33 and #2011.311). All study participants provided written informed consent for their participation. 

### 2.2. Sample Collection

Peripheral blood samples were collected pre-operatively after at least 8 h fasting. Whole blood was centrifuged at 2000× *g* for 10 min to separate plasma and blood cells. Plasma was aliquoted and stored in −80 °C freezers until analyses. DNA was extracted by the salting-out method from blood cells and stored in −20 °C freezers before genotyping [16]. 

Cartilage samples were obtained from the femoral head during TJR surgery and then flash-frozen and stored in liquid nitrogen. 150~200 mg frozen cartilage sample was transferred to the homogenizing cylinder together with 1 mL TRIzol Reagent (Thermo Fisher, Waltham, MA, USA) and 150 μL guanidine thiocyanate (Sigma-Aldrich, St. Louis, MO, USA) and homogenized in LN_2_ using a cryogenic mill (Spex Freezer Mill, model 6770, Metuchen, NJ, USA) with the following procedure: 10 min pre-cooling, and then 3 cycles of 1 min grinding at maximum frequency with 3 min for cooling between grinding cycles. The sample was then transferred to a 50 mL centrifuge tube and thawed at room temperature (RT), and then incubated for another 5 min after the sample had reached RT. 250 μL of chloroform (Thermo Fisher, Waltham, MA, USA) was added to the homogenate which was then transferred to a new 2 mL RNase-free tube and shaken vigorously for 15 s to mix thoroughly. The mixture was incubated at RT for 2~3 min and then centrifuged at 12,000× *g* for 15 min at 4 °C. After centrifugation, the sample was separated into 3 phases: the aqueous phase containing RNA, and the interphase and organic phase containing DNA. The aqueous phase was carefully transferred to a new 2 mL RNase-free tube and used for extracting total RNA by RNeasy Mini Kit (Qiagen, Hilden, Germany) following the manufacturer’s standard protocol and stored in −80 °C freezers until experiments.

### 2.3. Data Collection

Demographic data were obtained from general health questionnaires and medical records, age at surgery was calculated, and body mass index (BMI) was calculated as weight in kilograms divided by squared height in meters. Data on comorbidities including diabetes, hypertension, cardiovascular diseases, and other relevant conditions were obtained by self-reported questionnaires. 

Metabolomic profiling on fasting plasma samples was performed using Biocrates MxP Quant 500 kit (BIOCRATES Life Sciences AG, Innsbruck, Austria), which quantifies up to 630 different endogenous metabolites (a full list of the metabolites is provided in Appendix A). MxP Quant 500 kit contains a 96 deep-well plate with a filter plate attached with sealing tape, and reagents and solvents used to prepare the plate assay. Plasma was thawed on ice and samples were vortexed and centrifuged at 13,000× *g*. The first 14 wells are allocated to a blank sample, three zero samples, seven-point calibration standards, and three quality control samples (plasma with known metabolite concentrations at three levels). 10 µL of each plasma sample was loaded onto the remaining wells of the 96-well kit plate (the order of samples was randomized to ensure that the results were not affected by the plate well order) and dried in a stream of nitrogen. Subsequently, 20 µL of a 5% solution of phenyl-isothiocyanate was added for derivatization. After incubation, the filter spots were dried again using an evaporator (Glas-Col, Terre Haute, IN, USA). Extraction of the metabolites was then achieved by adding 300 µL methanol containing 5 mM ammonium acetate. The extracts were obtained by centrifugation into the lower 96-deep well plate, followed by a dilution step with kit MS running solvent. 150 µL of the extract was diluted with 150 µL water for LC-MS/MS analysis and 10 µL of the extract was diluted with 490 µL FIA solvent for FIA-MS/MS analysis. 

Mass spectrometric analysis was conducted using an Agilent 1290 UHPLC (Agilent, Santa Clara, CA, USA) coupled to Sciex QTrap 5500 Mass spectrometer (Sciex, MA, USA) equipped with an electrospray ionization unit. The analysis was run in multiple reaction monitoring mode, with data collected using Sciex Analyst software and peak review and analytical quantitation performed using MetIDQ software (BIOCRATES Life Sciences AG, Innsbruck, Austria). MetIDQ software was used to control the entire assay workflow from sample registration and automated calculation of metabolite concentrations to the export of data into other data analysis programs. Metabolite concentrations were reported as μMol/L. The analysis was completed at The Metabolomic Innovation Centre (TIMC; https://metabolomicscentre.ca (accessed on 2 January 2023)). 

Levels of pro-inflammatory cytokines: macrophage migration inhibitory factor (MIF), tumor necrosis factor-α (TNF-α), interleukin-6 (IL-6), and interleukin-1β (IL-1β) were measured in duplicates using sandwich ELISA kits (Human MIF DuoSet ELISA; DuoSet ELISA Ancillary Reagent Kit 2; Human TNF-alpha Quantikine HS ELISA; Human IL-6 Quantikine HS ELISA Kit; Human IL-1 beta Quantikine HS ELISA Kit, R&D systems, Minneapolis, MN, USA). All assays were performed according to the manufacturer’s instructions [15]. 

Genome-wide genotyping was done with either Illumina Omni-2.5 or Global Diversity Array genotyping platforms (Illumina, San Diego, CA, USA) and then imputed genome-wide using 1000 G phase 3 data as reference panels by the Sanger Imputation Service (https://imputation.sanger.ac.uk (accessed on 11 September 2022)) which generated data on a total of 81,706,022 genetic variants. 

The quality control (QC) filtering of RNA samples was performed using BioAnalyzer or TapeStation (Agilent Technologies, Waldbronn, Germany). Following the criteria set by Genome Québec (https://www.genomequebec.com (accessed on 6 April 2023)), only RNA samples with concentrations > 30 ng/μL and RNA integrity number (RIN) > 6.5 were used for RNA-sequencing. The stranded sequencing library was prepared using NEB Directional RNA Library Prep Kit for Illumina (New England Biolabs, Ipswich, MA, USA). RNA was sequenced using the Illumina NovaSeq 6000 S4 PE100 platform (Illumina, San Diego, CA, USA). Raw read counts were normalized by the counts per million (CPM) method and used in the subsequent analysis [17]. RNA samples with concentrations >65 ng/μL and RNA integrity number equivalent (RINe) > 7 were profiled using Affymatrix Human Clariom D Array (Affymatrix, Santa Clara, CA, USA) at The Centre for Applied Genomics (https://www.tcag.ca (accessed on 17 August 2023)). Three samples were assayed using both methods to assess the reliability of combining results from the two methods. 

### 2.4. Statistical Analysis

For the metabolomics analysis, metabolites were removed if more than 25% of the samples had values below the limit of detection (LOD) [18]. Values < LOD for the remaining metabolites were imputed using a linear regression prediction model in which concentration of a given metabolite was regressed on age, sex, and BMI. Principal component analysis (PCA) was performed, and no batch effect was detected (Appendix A). Considering the physiological variations in metabolite concentrations in the blood, which may differ by multiple orders of magnitude between individuals, pareto scaling was used to reduce the influence of data with high magnitude while maintaining integrity of dataset at the same time. Pareto scaling balances variable influence in multivariate analyses by scaling them with the square root of their variance, providing data pre-processing for metabolomics analysis [19]. The clustering analysis was done with the Uniform Manifold Approximation and Projection (UMAP) which reduces the dimensions of the metabolomic data to two dimensions, and visualizes and classifies study participants into different clusters clearly. The traditional clustering method K-means was used to validate the clustering result. The elbow plot was used to determine the optimal number of clusters, and K-means was applied on original data. The K-means clustering result was mapped to the UMAP clustering results to evaluate the reliability of UMAP.

Student’s *t*-test or Chi-squared test was used to compare the demographic variables. Pairwise Student’s T-test was performed between the identified clusters to identify the significant metabolites between clusters. Logistic regression and receiver operating characteristic (ROC) curve were then used to select the optimal prediction combinations and evaluate their prediction performances. Bonferroni correction was used to control for multiple testing and avoid false positives for identification of important metabolites. For the genome-wide association studies (GWAS), QC procedures were applied on the imputed GWAS data before association testing was conducted. All the genetic variants with minor allele frequency (MAF) < 0.01, genotype missingness > 0.03, and Hardy–Weinberg Equilibrium (HWE) test *p* < 0.0001 were removed. GWAS was used to identify the genetic variants for the identified metabolites that were associated with hip OA clusters. The top ten principal components were included in the model to control for population stratification [20]. snpXplorer was used to map SNPs into corresponding genes [21]. For transcriptomic data, genes expressed in less than 80% of the samples were removed, and the Wilcoxon signed-rank test was used to compare the gene expression levels between OA-affected cartilage and OA-free cartilage. MetaboAnalyst 5.0 (https://www.metaboanalyst.ca (accessed on 8 January 2024)) was utilized for the enrichment of metabolites into metabolic pathways [22]. Gene Ontology (GO) and Kyoto Encyclopedia of Genes and Genomes (KEGG) analyses were used for functional and pathway enrichment analyses.

All statistical analyses were performed in R version 4.3.2 with umap [23], dplyr [24], ROCit [25], ggplot2 [26], CMplot [27], ramwas [28], clusterProfiler [29], and PLINK 1.90 beta [30].

## 3. Results

### 3.1. Participants’ Characteristics

A total of 180 hip OA patients (age: 66.9 ± 9.5 yrs; BMI: 31.1 ± 6.2 kg/m^2^; 50.5% females) who underwent TJR surgery due to primary hip OA and 120 OA-free controls (age: 56.4 ± 8.8 yrs; BMI: 29.5 ± 4.8 kg/m^2^; 59.2% females) were included in the clustering analysis. Hip OA patients were significantly older and had a higher BMI than controls (*p* = 2.1 × 10^−19^ and 0.02, respectively), with no difference in sex distribution (*p* = 0.178). More details are presented in Table 1.

### 3.2. Clustering of Participants by Metabolomic Data

Six hundred and twenty-two out of the 630 metabolites passed the QC and were included in the analysis. Initial screening was performed by individual metabolite comparison between hip OA patients and OA-free controls using Student’s *t*-test; this identified 212 metabolites whose concentrations were significantly different between two groups with *p* < 0.05 (Appendix A). The UMAP analysis based on the 212 metabolites identified three clearly separated clusters (Figure 1). G0 cluster included the majority of the controls (n = 115) and a small number of hip OA patients (n = 19); G1 cluster contained 5 controls and 105 hip OA patients; G2 cluster contained 56 hip OA patients. Thus, we considered G0 as the non-OA cluster, and G1 and G2 as two separate hip OA endotypes. This clustering had a sensitivity, specificity, and accuracy of 95.8%, 89.4%, and 92.0%, respectively, to distinguish hip OA patients from controls. The robustness of UMAP clustering results was validated using the traditional K-means clustering method and data splitting. The elbow plot estimated that 300 participants could be divided into three main groups based on metabolomic data (Appendix A). The K-means clustering analysis based on the original metabolomic data classified 257 out of 300 individuals into the same clusters as identified by the UMAP analysis (Appendix A). Furthermore, we randomly split the data into subsets of 90%, 80%, 70%, and 60%, and all the subsets showed similar clustering patterns in the UMAP analyses (Appendix A). 

Pairwise Student’s *t*-test identified 27 significant metabolites between G0 and G1, 57 between G0 and G2, and 40 between G1 and G2 (all *p* < 8 × 10^−5^ which was Bonferroni corrected significance level after controlling 622 tests) (Appendix A). These metabolites were subsequently enriched into 8, 10, and 13 pathways, respectively, using MetaboAnalyst 5.0 (Appendix A). Five pathways were commonly shared among all the groups, consisting of arginine and proline metabolism, glutathione metabolism, arginine biosynthesis, cysteine and methionine metabolism, and glycerophospholipid metabolism. 20 significant metabolites were involved in these common pathways, and nine (arginine, ornithine, seven lysophosphatidylcholines (lysoPCs)) of them remained significant after adjustment for age, sex, and BMI. The combination of arginine, ornithine, and the average value of seven lysoPCs had an area under the curve (AUC) of 0.97 (95% CI: 0.96–0.99) to discriminate hip OA from OA-free controls. To investigate the uniquely altered pathways between G1 and G2, five common pathways were excluded from the initially identified thirteen pathways. 11 metabolites were involved in the remaining eight pathways, four (GABA (γ-aminobutyric acid), spermine, AconAcid (aconitic acid), and Suc (succinic acid)) remained significant after adjustment for age, sex, and BMI which had an AUC of 0.96 (95% CI: 0.94–0.99) to distinguish G1 from G2 (Figure 2). 

### 3.3. Genomic and Transcriptomic Analysis of Identified Metabolites

All NFOAS participants with both genomic and metabolomic data were included in the GWAS analysis to increase the statistical power for identifying SNPs associated with the identified metabolites. In total, 673 participants (including 300 who were part of a clustering analysis) had both types of data available for the GWAS. This group included 375 women (55.7%), with an age of 64.7 ± 9.3 years and an BMI of 32.9 ± 6.7. 9,612,957 autosomal variants were included in the GWAS analysis. GWAS identified 236 SNPs with a minor allele frequency > 1% to be associated with identified metabolites at the GWAS significance level (*p* < 5 × 10^−8^) (Appendix A). Quantile-quantile (QQ) plots of metabolite GWAS analyses (Appendix A) showed that the inflation factor (λ) ranged from 0.998 to 1.025, indicating the absence of population stratification. 

Subsequently, these SNPs were annotated to 130 genes (details of the 236 SNPs and the annotation result are provided in Appendix A). To further investigate our findings for the 130 GWAS significant genes, we explored gene expression data in hip cartilage. Not all the participants have available cartilage to measure due to being at end-stage OA; therefore, the transcriptomic analysis only included a subset of participants. However, these patients did not significantly differ in age, sex, or BMI from those without available cartilage (*p* = 0.85, 0.15, and 0.75, respectively). The transcriptomic data consisted of 12 OA-free cartilage (age: 82.8 ± 10.0 yrs; BMI: 23.3 ± 4.9 kg/m^2^; 75.0% females) and 72 hip OA-affected cartilage (age: 67.8 ± 10.6 yrs; BMI: 30.8 ± 6.2 kg/m^2^; 55.6% females). 36 cartilage RNA samples were sequenced with RNA-Seq and 51 were profiled with the microarray method, and expression levels were scaled by the Z-score method. The three samples analyzed using both methods showed excellent concordance between the two datasets. Eleven genes were not detected in cartilage, 27 genes were excluded due to the absence of expression in more than 20% of the samples, and the remaining 92 genes were included in the analysis. We found that the expression levels of 10 genes (*C12orf75*, *TMEM117*, *ADCY1*, *NUAK1*, *SH3D19*, *TENM3*, *STXBP6*, *DCLK2*, *ZNF503*, and *METTL2B*) were significantly higher in OA-affected cartilage than in OA-free cartilage, while the expression levels of four genes (*MARCHF3*, *TGIF1*, *HILPDA*, and *ZNF385D*) were lower (*p* < 5 × 10^−4^ after adjusting for 92 genes tested). In addition, GO and KEGG enrichment analyses identified four pathways associated with inflammation, including response to interleukin-1, cytokine activity, the HIF-1 signaling pathway, and the TNF signaling pathway, as well as two pathways associated with energy metabolism, including purine metabolism and the GMP metabolic process. (Appendix A).

### 3.4. Pro-Inflammatory Cytokines Measurement

A total of 77 participants had pro-inflammatory cytokine data available; among them, 22 were classified into G0, 29 into G1, and 26 into G3. The G0 (5/22 females) had an average age of 63.6 ± 9.1 years with a mean BMI of 28.7 ± 3.5 kg/m^2^, the G1 (9/29 females) had an average age of 65.5 ± 10.7 years with a mean BMI of 29.2 ± 5.0 kg/m^2^, and the G2 (10/26 females) had an average age of 64.8 ± 8.7 years with a mean BMI of 30.2 ± 3.9 kg/m^2^. There were no significant differences in age, sex, and BMI among the three clusters (all *p* > 0.05). Among the four measured pro-inflammatory cytokines, three (MIF, TNF-α, and IL-1β) of them showed significant differences between the three clusters (all *p* < 0.05) (Table 2). Additionally, these three cytokines were significantly correlated with seven identified metabolites, with coefficients ranging from −0.39 to 0.84 (Figure 3). 

124 out of 180 patients completed the questionnaire (n = 80 and 43 for G1 and G2, respectively). The missing values in G0 were due to the fact that the participants classified as G0 were mainly from the Complex Diseases in the Newfoundland Population: Environment and Genetics (CODING) study, which did not have data on these conditions.

Cardiovascular diseases included coronary heart disease, heart attack, hypertension, angina, high cholesterol, deep vein thrombosis, varicose veins, and pulmonary embolism; Immunology included asthma, hayfever, eczema, sinusitis, and Crohn’s; Gastroenterology included heartburn and irritable bowel syndrome; Neurological/Psychiatric diseases included clinical depression, anxiety/stress disorder, epilepsy, stroke, motion sickness, and migraine; Rheumatological diseases included gout, Paget’s disease, bunions, frozen shoulder, osteoporosis, carpal tunnel, and tennis/Golfer’s Elbow.

Family history referred to immediate family’s OA history (only father and mother included). If at least one parent of the participant has been diagnosed with OA, the participant’s family history was recorded as family history (+).

Sustained pain was defined as patient self-reporting at least one point in each of the five questions in Western Ontario and McMaster Universities Osteoarthritis Index Likert 3.0 pain subscale. 

77 patients had available pro-inflammatory cytokines data; among them, 22 were classified into G0, 29 into G1, and 26 into G3.

## 4. Discussion

In the current study, we identified two biomarker panels that can distinguish OA-free individuals from OA patients and two endotypes of hip OA, respectively. Interestingly, all seven identified metabolites were related to the tricarboxylic acid (TCA) cycle and the urea cycle.

The panel distinguishing hip OA patients from controls consisted of arginine, ornithine, and the average value of seven lysoPCs, with an AUC of 0.97. Our results suggested that increased synthesis of lysoPCs and enhanced conversion pathway from arginine to ornithine may contribute to the pathogenesis of hip OA. LysoPCs are mainly generated from phosphatidylcholines (PC) by enzyme phospholipase A2, and fatty acids are released in tissues during this process [31]. LysoPCs have been reported to play a crucial role in various signaling pathways associated with oxidative stress, inflammatory responses, and apoptosis [32]. Moreover, the released fatty acids are precursors to bioactive molecules such as prostaglandins and leukotrienes, which are involved in multiple pathways including inflammatory responses [33]. For the other two components, a previous study conducted by our lab has shown that arginine levels were significantly lower and ornithine levels were significantly higher in patients with knee OA compared with healthy controls [34]. In the current study, similar dysregulation patterns were observed in hip OA patients. Notably, there appeared to be a more extensive conversion of arginine to ornithine in the urea cycle in G2 compared to G1. Nitric oxide (NO) is released during the conversion of arginine to citrulline [35]. NO has been widely reported to contribute to the progression of OA by promoting the expression of pro-inflammatory cytokines such as IL-1β, TNF-α, and NF-κB, inhibiting collagen synthesis, and inducing apoptosis in cartilage [36]. Briefly, our findings indicated that two pathways were involved in the development of hip OA, which primarily involved inflammation and apoptosis in cartilage. 

The panel separating two hip OA endotypes consisted of GABA, spermine, AconAcid, and Suc, of which three were involved in the TCA cycle and its GABA shunt. The concentrations of all these biomarkers were significantly higher in G2 than in G1. As mentioned previously, G2 had higher ornithine and lower arginine levels compared with G1. Arginine is synthesized from citrulline by the sequential action of the argininosuccinate synthetase and argininosuccinate lyase in kidney [35]. Compared to G0, citrulline level was significantly higher in G2 but showed no difference in G1, indicating that lower arginine in G2 was not purely due to impaired endogenous synthesis. Furthermore, the conversion of citrulline to arginine in G2 might be enhanced due to the compensatory response, while more fumarate was released to participate in the TCA cycle. Therefore, the lower arginine level in G2 could be attributed to an enhanced pathway that converts arginine to ornithine. Meanwhile, ornithine is converted to spermine through a series of enzymatic reactions [37]. Significantly higher concentrations of ornithine, spermidine, and spermine in G2 compared to G1 were observed, suggesting that the arginine to ornithine to spermine pathway was overactive in G2. Spermine can suppress the synthesis of proinflammatory cytokines in monocytes by restraining macrophages [38]. In addition, it has been reported that spermine and spermidine can ameliorate cartilage and bone destruction in synovial joints and reduce synovitis, cartilage degeneration, and osteophyte formation in a rat model [39,40]. In the TCA cycle and its GABA shunt, three identified metabolites were highlighted (Figure 4). Mitochondrial aconitase facilitates the conversion of citrate to isocitrate, with AconAcid as the intermediate [41]; α-ketoglutarate is converted to succinyl-CoA, which is then synthesized into Suc and one molecule of GTP [42]; additionally, α-ketoglutarate is also involved in the GABA shunt. It is converted to GABA through several enzymatic reactions, GABA is then metabolized by GABA transaminase and succinic semialdehyde dehydrogenase, allowing it to re-enter the cycle, thereby completing the loop [43]. AconAcid has been reported to act as a cardiovascular risk factor [44], however, there was no difference in cardiovascular disease prevalence between G1 and G2 in the current study. Although mild positive correlations (correlation coefficient: 0.47 and 0.43, respectively) between AconAcid and the pro-inflammatory cytokines IL-1β and TNF-α were found in our data, a study reported that AconAcid can reduce inflammation by inhibiting the levels of IL-1β and TNF-α in arthritic mice [45]. Suc level rises during inflammation, activating macrophages to produce hypoxia-inducible factor-1α (HIF-1α), which subsequently sustains IL-1β production [46]. In addition, anoxic succinate has been shown to activate NLRP3 inflammasome through HIF-1α in an arthritic rat model, thereby exacerbating synovial fibrosis and inflammation [47]. Another study conducted on arthritic rats found that accumulation of Suc led to GPR91-dependent succinate cycling, subsequently increasing IL-1β levels and exacerbating inflammation [48]. Similar to the observations with AconAcid, positive correlations between GABA and three pro-inflammatory cytokines were also found, which was inconsistent with a previous study demonstrating that GABA treatment inhibited IL-1β production by inflammatory macrophages in vitro [49]. Although the cingulate GABA has been reported to be negatively associated with sustained pain in OA patients [50], our data did not show a significant difference in Western Ontario and McMaster Universities Arthritis Index (WOMAC) pain scores between G1 and G2. 

The components in the panel distinguishing G1 and G2 exhibited an opposite function in inflammation. Some of these components have been shown to have anti-inflammatory functions in previous studies, while others are thought to have pro-inflammatory effects. However, all the components showed significant positive correlations with pro-inflammatory cytokines in the current study. Considering that pro-inflammatory cytokines were upregulated in G2, the high levels of spermine, AconAcid, and GABA in G2 could be attributed to an unbalanced biological process that is attempting to inhibit inflammation in G2. Therefore, G2 is defined as an endotype of hip OA characterized by high inflammation.

Southan J. et al. observed decreased levels of arginine and increased levels of ornithine in cartilage metabolomic analysis in an animal model of hip cartilage injury [51]. Kosai A. et al. found that blocking the TCA cycle can delay chondrocyte maturation [52]. This similar pattern of changes suggests that metabolic disturbances in plasma may indicate potential changes in cartilage health and function. This provided further evidence of the potential of using plasma metabolites as biomarkers for diagnosis and classification of OA, as we previously demonstrated [53].

In GWAS analysis, 236 SNPs were identified to be associated with the identified metabolites. rs188452133-A in *ABCB5* was associated with succinic acid. *ABCB5*-knockout melanoma cells have lower succinic acid levels than wild-type cells [54]. *ABCB5* is an ATP-dependent efflux transporter, and its expression may lead to increased energy demand, thereby altering the balance of the TCA cycle. rs148800533-T in *ANGPT1* is associated with GABA, and abnormal expression (overexpression or deficiency) of *ANGPT1* leads to a decrease in GABAergic neurons, which can produce GABA [55]. Furthermore, we identified 14 genes which had significant differences between OA-affected cartilage and OA-free cartilage. Among these upregulated genes, the most significantly upregulated gene, *C12orf75*, has been found to be associated with insulin signaling, energy metabolism, and the aging of skeletal muscle [56]; *ADCY* catalyzes the conversion of ATP to cAMP, and then cAMP acts as a signal to activate the expression of genes related to glucose metabolism [57]; *NUAK1* negatively regulates insulin signaling in skeletal muscle, thus controlling glucose metabolism [58]; *STXBP6* knockout mice showed leaner body weight and higher expression of genes associated with negative regulation of peptidase activity than wild-type mice [59]. Moreover, the downregulated gene *MARCH3* has been reported to exacerbate the expression of pro-inflammatory cytokines in *MARCH3*-deficient mice [60]. In osteoblasts from *TGIF1* knockout mice, the loss of *TGIF1* was found to reduce alkaline phosphatase (ALP) activity [61]. ALP can promote bone mineralization and enhance fat cell metabolism [62]. *HILPDA* deficiency in knockout mice was shown to promote adipose tissue lipolysis, leading to enhanced lipid droplet degradation and promoting fatty acid oxidation in mitochondria [63]. *ZNF385D* was associated with increased levels of ceramide (d18:1/24:0) in the blood, but no significant differences in this metabolite were observed in our cohort (*p* = 0.60) [64]. *ZNF385D* may be involved in lipid metabolism in some unknown pathway, which needs further investigation. These metabolite-related genes were directly or indirectly involved in lipid metabolism or energy metabolism through various pathways, suggesting that some metabolic pattern was changed in hip OA.

Overall, our results suggested that the common altered pathways in two hip OA endotypes might involve inflammation and apoptosis in cartilage. G1 was considered as an endotype characterized by relatively moderate inflammation with fewer comorbidities, while G2 was an endotype featuring more severe inflammation and alteration in energy metabolism. Age, sex, and BMI may affect energy metabolism and pro-inflammatory cytokines levels [65]. However, there were no differences in these demographic factors between the two endotypes (Table 2), suggesting that the alterations in energy metabolism pathways and inflammatory levels were not attributable to demographic factors.

There were some limitations to this study. First, all the OA patients in the current study underwent THR, which indicated they were at the end stage of the disease. Hence, the findings may not apply to early stages of the disease. Secondly, the existing comorbidities and the significant differences in age and BMI between participants could affect metabolites, cytokines, and gene expression levels. Although the confounding effects were adjusted, the application of our findings to younger patients or those with normal BMI needs to be tested. Thus, an independent cohort with large sample size is needed to validate our findings. Thirdly, gene expression and cytokine data were not available in all study participants, thus potential bias might exist. Lastly, this was a cross-sectional study, hence the causal relationships between identified markers and hip OA remain to be investigated. 

## 5. Conclusions

In summary, our data demonstrated the existence of two distinct endotypes in hip OA, which can be identified using a unique combination of blood metabolic markers. Although further studies are needed to confirm their validity and generalizability, our findings have a great potential in developing personalized tools for clinical hip OA management and targeted care for different endotypes of hip OA. Biologicals targeting the pro-inflammatory cytokines examined in the current study are already on the market, thus, our findings may help to develop endotype-based biological therapeutic strategy for hip OA patients. 

## Figures and Tables

**Figure 1 metabolites-14-00480-f001:**
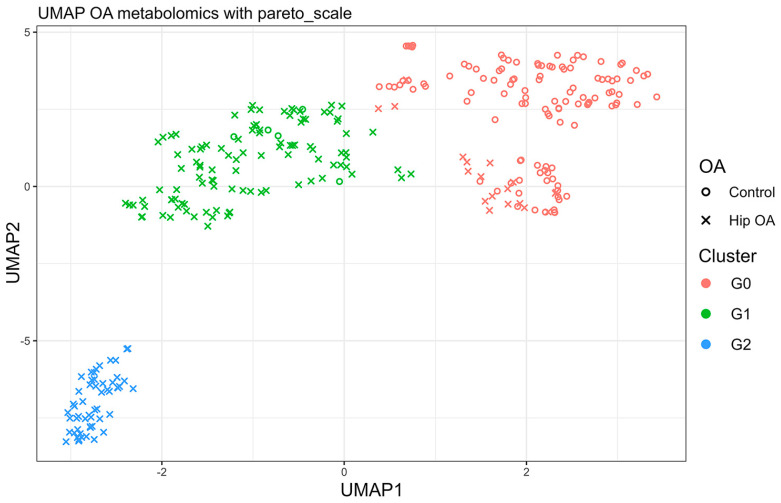
Clustering analysis result. Uniform Manifold Approximation and Projection (UMAP) clustering results based on concentrations of 212 metabolites that were associated with hip OA with *p* < 0.05. The 300 individuals were classified into three separate groups marked by different colors.

**Figure 2 metabolites-14-00480-f002:**
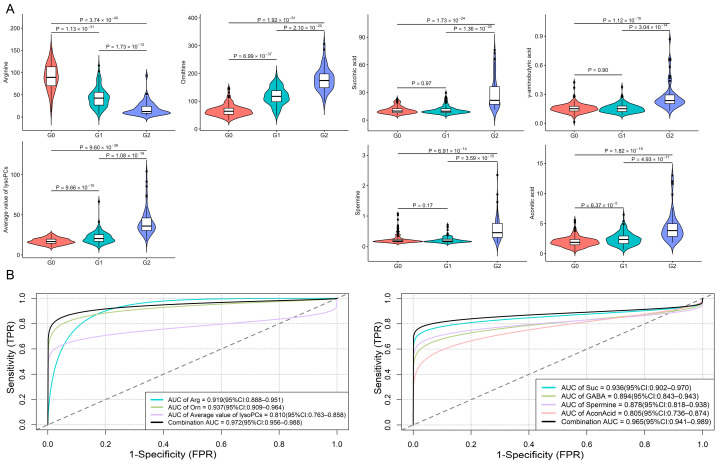
Individual metabolite concentrations and discrimination performance. (**A**) Violin plots show the concentrations of the components of the diagnostic panel used to distinguish healthy controls from hip OA at G0, G1, and G2. The statistical significance was calculated by Student’s *t*-test. Violin plots show the concentrations of the components of the diagnostic panel used to distinguish two hip OA endotypes at G0, G1, and G2. (**B**) ROC curve analysis results for the two combinations for discriminating G0 vs. G1 + G2 and G1 vs. G2.

**Figure 3 metabolites-14-00480-f003:**
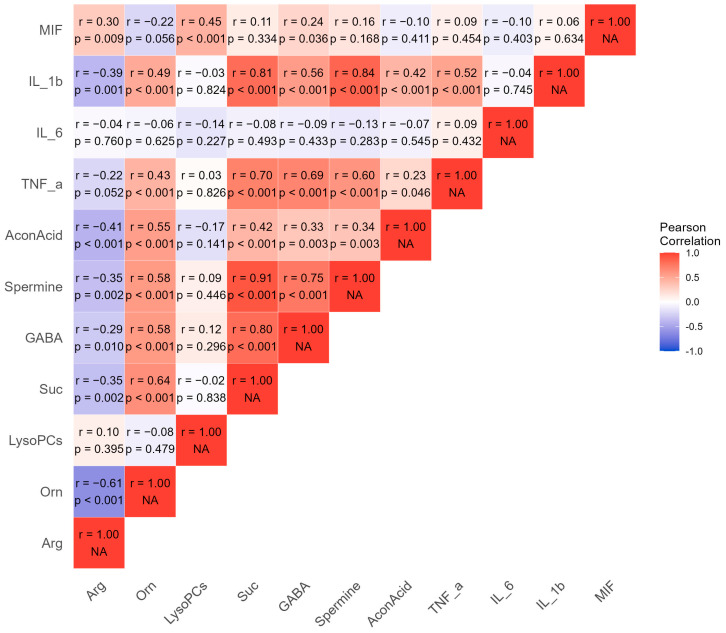
Correlation matrix of identified metabolites and pro-inflammatory cytokines. Pearson rank correlation matrix of identified metabolites and pro-inflammatory cytokines. Red represents a positive correlation, whereas blue represents a negative correlation. Arg: arginine; Orn: ornithine; LysoPCs: lysophosphatidylcholines; GABA: γ-aminobutyric acid; AconAcid: aconitic acid; TNF-α: tumor necrosis factor-α; IL-6: interleukin-6; IL-1β: interleukin-1β; MIF: macrophage migration inhibitory factor.

**Figure 4 metabolites-14-00480-f004:**
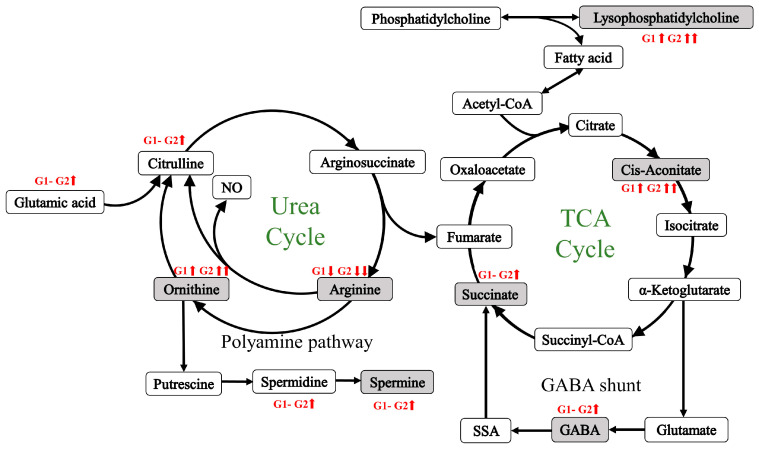
Metabolic pathway of the urea cycle and tricarboxylic acid cycle (TCA) and its γ-aminobutyric acid (GABA) shunt. The identified metabolites were marked with a gray background. Concentrations of metabolites measured in the current study in G1 and G2 (compared to G0) were indicated by arrows (red font).

**Table 1 metabolites-14-00480-t001:** Comparison of demographic factors between hip OA patients and OA-free controls.

	Hip OA	OA-Free Control	*p* Value
Age (years)	66.9 ± 9.5	56.4 ± 8.8	2.1 × 10^−19^
BMI (kg/m^2^)	31.1 ± 6.2	29.5 ± 4.8	0.02
Sex (female%)	50.5%	59.2%	0.178

Values are either mean ± standard deviation or percentage. *p*-values were obtained by Student’s *t*-test (age, BMI) or Chi-squared test (sex).

**Table 2 metabolites-14-00480-t002:** Patients’ characteristics and clinical data of three clusters.

	G0 (n = 134)	G1 (n = 110)	G2 (n = 56)	*p* ^#^	*p* ^*^
Age (years)	58.2 ± 9.9	67.0 ± 10.1	65.0 ±8.6	2.00 × 10^−11^	0.18
BMI (kg/m^2^)	29.8 ± 5.0	30.6 ± 6.4	31.8 ± 6.1	0.09	0.24
Sex (female%)	54.4%	50.9%	58.9%	0.61	0.42
Diabetes (%)	31.3%	8.1%	10.7%	5.76 × 10^−6^	0.80
Cardiovascular diseases (%)	-	11.3%	13.2%	-	0.80
Immunology-related diseases (%)	-	6.3%	3.3%	-	0.12
Gastroenterological diseases (%)	-	18.7%	18.6%	-	0.97
Neurological/Psychiatric diseases (%)	-	3.8%	7.0%	-	0.31
Rheumatological diseases (%)	-	5%	9.6%	-	0.25
OA Family History (%)	-	58.8%	44.1%	-	0.12
Sustained pain	-	6/73	4/39	-	0.71
MIF	8.3 ± 8.1	4.2 ± 2.3	5.6 ± 1.7	9.60 × 10^−3^	0.13
TNF-α	1.3 ± 0.4	1.1 ± 0.3	2.0 ± 1.5	1.04 × 10^−3^	4.80 × 10^−3^
IL-1β	20.2 ± 14.7	33.6 ± 20.3	55.3 ± 7.4	2.1 × 10^−7^	3.00 × 10^−4^
IL-6	35 ± 21.2	39.4 ± 23.2	38.7 ± 20.7	0.68	0.52

Values are either mean ± standard deviation or percentage. *p*^#^: statistical differences between three clusters using ANOVA. *p*^*^ statistical differences between G1 and G2 clusters using Student’s *t*-test or Chi-squared test.

## Data Availability

Data are available upon reasonable request from the corresponding author.

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
