# Peer review of "Multi-Omics Integrative Analyses Identified Two Endotypes of Hip Osteoarthritis"

_metabolites, 2024, doi:10.3390/metabo14090480_

Round 1

Reviewer 1 Report

Comments and Suggestions for Authors

This manuscript describes a multi-omics approach to develop endotypes of hip osteoarthitis.  The authors have a large cohort of plasma samples from both normal and OA patients, as well as some sequencing data from both whole blood and cartilage.  There may be merit in this data, but there are several flaws in analysis that must be corrected.

Major concerns include:

1.     For the metabolite comparisons, t-tests were used on each of the more than 600 metabolites detected using the Biocrates kit.  However, no multiple comparison corrections were used—multiple comparison corrections are needed when testing large numbers of hypotheses as done here.

2.     There is no validation cohort.  The authors should test a second group of samples (or pull data from their multiple prior studies on hip OA) and see if the same clustering approach replicates blusters G1 and G2

3.     Per Table 1, there are significant differences in both age and BMI between the OA and control groups.  This is not mentioned as a limitation, and these differences likely affected the results.

4.     Line 171, more explicit description of the methods that used k-means to validate the initial clustering are required

5.     There is no rationale presented for the selected sample sizes.

6.     In general, much of the text in the figures (axis labels, etc) is very small and difficult to read. 

7.     Readers would benefit from plotting the data of Figure 2 as a violin plot

8.     Figure 3 needs a label on the colorscale bar.

9.     The Discussion could benefit from text describing the relationship between cartilage and plasma metabolites.

Comments on the Quality of English Language

There are some grammar and wording improvements that would help readability.

Reviewer 2 Report

Comments and Suggestions for Authors

Major revisions

1.      The  description of state of the art should be improved; it would be appropriate to rewrite the introduction to provide additional details about the OA and its classification.

2.      The criteria for inclusion and exclusions should be identified in the materials and methods section for participants. Have clinical evaluations not been carried out or shown?

3.      The description of the quantitative mass analysis is poorly detailed, and the QC parameter is not shown.

4.        How was the batch effect evaluated? Provide additional graphic evidence in the material to support the absence of a batch effect

5.      How was the K parameter evaluated for implementing the K-means algorithm?  It would be appropriate to show the WCSS graph.

6.      The bioinformatics strategy underlying metabolomic analysis is unclear. It appears that a T-test was performed to reduce the variables, and then a UMAP approach was applied. Would it be possible to explain the choice or to describe more clearly how the analysis was carried out?

7.      It appears that the different analyses were not carried out on the same sample number. Why?

8.      The study is multi-omic, but the information is poorly integrated; the authors use machine learning algorithms but never integrate omics. Consideration should be given to the possibility of an integrated analysis.

9.        The correlogram shall be supplemented by the value of the coefficient obtained and the value of significance

10.  Remodulate the discussion by bringing out more of the biological correlation between the data of genomics and those of metabolomics also through a final bullet list or a division into paragraphs of the discussion itself

Comments on the Quality of English Language

The English language would require a moderate improvement.

Round 2

Reviewer 1 Report

Comments and Suggestions for Authors

I noticed another error in this paper in that it appears that there is an unneeded data imputation (for metabolites where <25% of the values are below threshold.) 

Comments on the Quality of English Language

Acceptable

Reviewer 2 Report

Comments and Suggestions for Authors

the autoris have replied to all requests
